# Measurement System for Finger Skin Displacement on a Textured Surface Using Index Matching

**Seitaro Kaneko [1,2,\*]**  **and Hiroyuki Kajimoto [2]**

1   Research Fellow of the Japan Society for the Promotion of Science, Japan Society for the Promotion of Science, 5-3-1 Kojimachi, Chiyoda-ku, Tokyo 102-0083, Japan
2   Department of Infomatics, The University of Electro-Communications 1-5-1 Chofugaoka, Chofu, Tokyo 182-8585, Japan; kajimoto@kaji-lab.jp
\*   Correspondence: kaneko@kaji-lab.jp; Tel.: +81-42-443-5445

**Abstract:** Understanding the relationship between the displacement of the skin when tracing a textured object and the resulting subjective sensations is essential in designing tactile displays. Previous studies observed skin displacement using flat glass plates or uneven surfaces that do not optically interfere with finger surface observations. In contrast, no direct method for observing skin surface displacement on a texture exists. We propose a system that enables observation of the interaction between a textured surface and the skin of the finger using an index-matching technique. In the proposed system, a texture plate is immersed in oil having the same refractive index as the plate, and measurements are made when the interface is nearly optically transparent. Further, printed markers are attached to the skin of the finger, and their movements analyzed using an image-processing algorithm. The system enables spatial measurement of the skin shear and the vibration of the contact area. Evaluation experiments conducted on a 1D textured surface having a pitch of 0.6 mm verify the feasibility of the proposed system. Optical misalignment simulation results indicate that the system is slightly less accurate than type-I mechanoreceptors but can measure skin deformation on a texture and also observe it spatially and temporally.

**Keywords:** tactile display; index matching; measurement system; textured surface; skin displacement

## 1. Introduction

Tactile displays that produce realistic sensations are used for virtual reality (VR), teleoperation, and remote palpation applications. Research seeking to enable more realistic simulated tactile presentations (i.e., the sense of friction or roughness) as felt by the fingertip is actively underway. The aim is to improve the sense of immersion in VR and remote control. Past skin displacement studies focused primarily on temporal changes. However, to present a high-quality tactile sensation, it is not enough to only measure temporal changes; it is also necessary to measure spatial changes, such as those of skin displacement.

In the past, accelerometers and force sensors have been relied upon to measure vibrations caused by friction [1] and fine textures [2], including artificial [3] and natural ones [4]. These measurements have focused on the activity of the slowly adapting type-II Pacinian corpuscle (PC) receptive fields, which fire relatively well at a combination of high frequency and low spatial resolutions. However, recent neurophysiological findings suggest that the slowly adapting type-I (SAI) mechanoreceptors (Meissner corpuscles) and the rapidly adapting type-I (RAI) mechanoreceptors (Merkels disk) are active at higher spatial resolutions when tracing textures [5]. Thus, in addition to temporal information, spatial information is clearly useful for reproducing vivid texture sensations of high quality.

Although there have been efforts to measure spatial skin displacement, previous studies did not measure sensations of roughness, such as that of a flat surface. Levesque et al. [6] measured skin displacement on flat or very large uneven surfaces (3 mm ridge width) using a flat plate and a high-speed camera. However, the pitch width was not psychologically related to roughness perception, but was instead related to shape perception [7]. Although it has been suggested that spatial skin changes contribute to texture perception, the spatial behaviors of roughness perception measurements, which are inherently spatiotemporal in nature, have not been directly observed. To enable this, a spatiotemporal haptic measurement system is needed.

To this end, we developed a measurement system for skin displacement on a wide range of textures that incorporates an index-matching method from the field of fluid dynamics. In the developed system, a transparent sample is submerged in oil with almost the same refractive index as the sample, and measurements are made when the interface is nearly optically transparent. In addition to 1D unevenness, this makes it possible to spatially measure the skin shear and vibration of the contact area when tracing a texture that accurately reproduces a natural acrylic texture sensation.

The contributions of this paper can be summarized as follows:

- We propose a method for directly observing skin displacement on coarse textured surfaces.
- The texture shapes to which this method can be adapted are clarified via optical simulation based on Snell's law.
- We verify the accuracy of the equipment that implements this method.

A limitation of this system is the change in friction caused by the use of oil, because finger-skin conditions are different from those of dry conditions. Friction has been reported to have an effect on roughness perception [8]. Thus, the perceived roughness is likely to be reduced for measurements in oil compared with normal measurements. It is also known that the mechanical properties of finger skin change with humidity. Dzidek et al. [9] showed that increasing fingertip humidity softens the fingertip belly and greatly affects the change in area when it comes into contact with an object. The data obtained by our measurement method are those of skin displacement in oil, which may differ from normal skin displacement. Thus, it may not be possible to reproduce the natural texture by simply reproducing the skin displacement in oil. However, even with oil constraints, it is useful to investigate the relationship between the subjective roughness sensation and skin displacement in contact areas.

## 2. Related Work

Tactile perception relies on multiple processes. These can be broadly divided into the conversion of object shape into skin vibration, the conversion of skin vibration into receptor-firing activity, and the processing of neural firing patterns in the brain. In this vein, neurophysiologists have attempted to measure the relationship between texture shape and receptor firing patterns. LaMotte et al. [10] and Srinivasan et al. [11] performed experiments to measure the neural activity of receptors in the fingertips of monkeys. They used simple textures, such as artificially created planes and steps, to record SAI and RAI temporal firing-pattern changes. Connor et al. [12,13] measured how fingertip receptors undergo spatiotemporal changes when presented with a matrix of dots. From this, they found a greater relationship between roughness and tactile receptor activity via spatial changes than via temporal changes at 1–2-mm dot intervals.

Skin vibrations have been recorded to reveal how texture shapes are decoded by the skin. Martinot et al. [2] used an accelerometer and a laser Doppler module to measure the similarity between vibrations and groove texture shapes when traced with a fingertip. Romano et al. [14] measured contact vibration data for a large number of natural textures using accelerometers, position sensors, and force sensors attached to pens and attempted to reproduce these results on a tactile display. Sato et al. [15] proposed a method of indirectly measuring the shear displacement of the finger surface using an accelerometer to measure changes on the finger side.

The behavior of spatial skin displacement has also been measured to elucidate patterns, such as stick-slip and shape perception. Numerous measurements have been made using transparent measurement samples and cameras. Levesque et al. [6] measured the behavior of the finger surface on flat and very large uneven surfaces after defining the feature points of the fingertips, including sweat pores and fingerprints. Soneda et al. [16] used a glass prism to measure the true and apparent contact areas based on the contact force. In another study, a similar optical measurement setup was used to measure changes in the moisture content and the contact area of the skin during the grasping of an object [17]. Kurita et al. [18] proposed a method to discriminate the force exerted on a fingertip and its orientation using the initial shear displacement of the fingertip contact area. Yuan et al. [19] proposed a method using GelSight tactile sensors to detect normal, shear, and torsional loads on contact surfaces on planar and curved surfaces. Such optical measurements of fingertip surfaces are not only used to reveal tactile perception but also for human–computer interfaces. Holz et al. [20] used a fingerprint image to identify which fingers were in contact with the display.

In summary, it can be said that, although the temporal changes on textures have been measured using accelerometers, fingertip behavior on very fine textures has not been directly and spatiotemporally measured. Although there have been clear examples of spatial measurements, most of the measured samples were of planar or large uneven shape that did not interfere with optical observations. The direct quantitative observation of spatiotemporal behavior is essential for presenting higher quality textures.

## 3. Measurement System

### 3.1. Principles

We use the index-matching method [21,22] from the field of hydrodynamics to enable direct skin displacement measurements on textures. Normally, when observing the skin surface through a texture, light is refracted at the textured surface because of the large difference in the refractive index between the plate and the air. This prevents a clear observation. The index-matching method is used to solve this problem. It is a technique for making a texture optically transparent by submerging a transparent object in a liquid that has the same refractive index. See Figure 1.

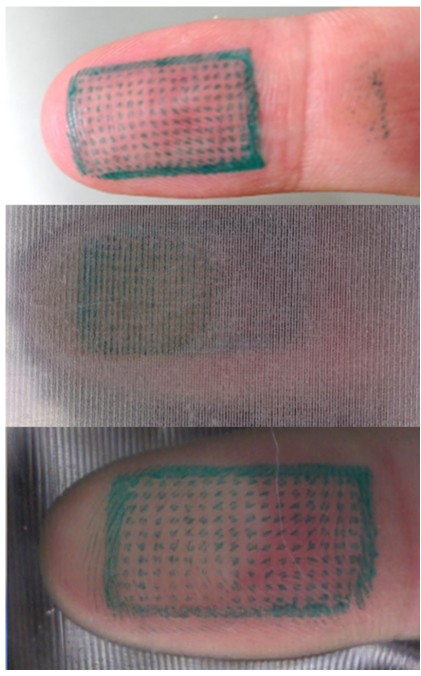

**Figure 1.** Top: fingertip with markers; middle: fingertip viewed through the textured surface without oil; bottom: fingertip viewed through the textured surface submerged in oil.

Using this method, it is very important to match the refractive indices of the sample and the medium, because when the refractive indices differ, optical deviations will occur. Here, we simulate the optical misalignment of the skin displacement measurement over a 1D triangular texture having three variables: the refractive index, angle of the triangular textured surface, and distance from the refractive point to the fingertip—defined as $n$, $\theta$, and $h$, respectively. In this system, an acrylic plate (refractive index: 1.490) is used. Figure 2 illustrates the angle of refraction used in the simulation. As temporary variables, the angle of incidence and the amount of optical deviation from the actual dot are set to $\varphi$ and $x$, respectively. Using the refractive-angle principle, the relation between the angle of incidence and the refractive angle is determined by the following equation based on Snell's law:

$$\varphi = \sin^{-1}\left(\sin\left(90° - \frac{\theta}{2}\right) \times \frac{1.49}{n}\right) \tag{1}$$

The optical misalignment, $x$, can be expressed as follows, using the angle of incidence from the dot $\varphi$, the angle of the texture $\theta$, and the distance to the fingertip $h$:

$$x = h \times \tan\left(90° - \frac{\theta}{2} - \varphi\right) \tag{2}$$

When the refractive indices are in perfect agreement ($n = 1.490$), no refraction occurs. Thus, the optical misalignment, $x$, is zero. Here, we estimate the degree of optical misalignment that occurs when we use silicone oil ($n = 1.485$) for the measurement system and water ($n = 1.333$) for comparison.

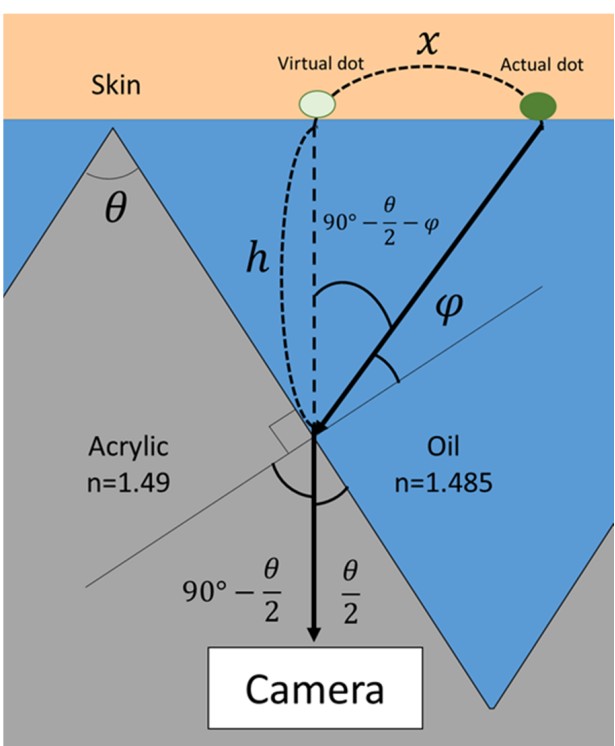

**Figure 2.** Overview of light refraction and the amount of optical misalignment caused by it. The refractive index of the medium, the angle of the texture, and the distance to the fingertip are defined as $n$, $\theta$, and $h$, respectively. As temporary variables, the angle of incidence and the amount of optical deviation from the actual dot are set to $\varphi$ and $x$, respectively.

Figure 3 shows the correspondence between the distance from the refractive point to the fingertip and the amount of optical misalignment. Figure 1 shows the correspondence between the distance from the refractive point to the fingertip and the amount of optical misalignment. Each line in the graph shows the amount of displacement when the texture surface angle, $\theta$, is changed. From each graph, it can be seen that the optical deviation becomes smaller when the refractive index is close to that of acrylic. The optical misalignment in silicone oil becomes smaller with smaller texture angles. When compared with a two-point discrimination threshold of 3 mm [23], which is considered to be the basic human spatial resolution, it is found to be very small.

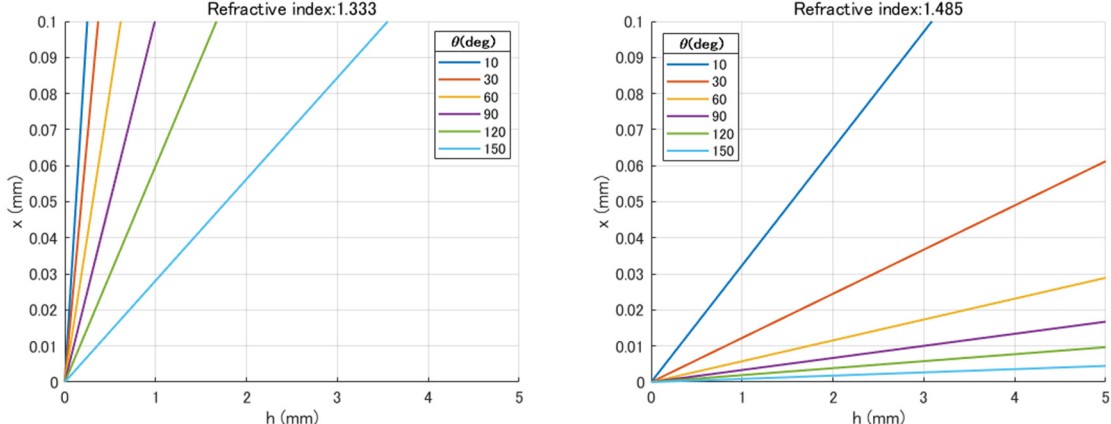

**Figure 3.** Relationship between distance from the refractive point to the skin (h) and optical misalignment (x) when using silicone oil (left) and water (right). Each line in the graph shows the relationship when the texture angle is changed.

Next, we compare the vibration detection thresholds of each receptor [24,25] (SAI: 100 µm; RAI: 10 µm; PCs: 1 µm) with the optical misalignment when silicone oil is used. When the height is less than 5 mm, and the texture angle is less than 30°, it is smaller than the threshold of SAI. Additionally, when the texture angle is large (about 90°), it is possible to measure the optical misalignment below the RAI threshold. Furthermore, it is difficult to measure the optical misalignment below the PC threshold.

The results show that it is possible to reduce the optical misalignment in silicone oil with a very close refractive index. It is also suggested that this measurement system principle is particularly effective for measuring skin displacement related to SAI and RAI receptor activity.

### 3.2. Hardware

Figures 4 and 5 show the experimental setup. Markers used for image processing are applied to fingertips using oil-resistant ink. The markers are arranged in a 10 × 16 arrangement with a diameter of 0.5 mm and a center-to-center distance of 1.0 mm (Figure 6). The markers are observed with a high-speed camera (SONY, RX10 II) capable of shooting at 1920 × 1080 pixels and 960 frames per second (fps). The camera observes a 57.6 × 32.4 mm region where one pixel corresponds to 30 µm. White light-emitting diodes (LED) are used for the fingertips to reduce the shooting noise caused by the high-speed camera. Acrylic plates are soaked in silicone oil having a refractive index of 1.485 (Shin-Etsu Silicone KF-53). The fingertips are also submerged in the liquid during the measurement.

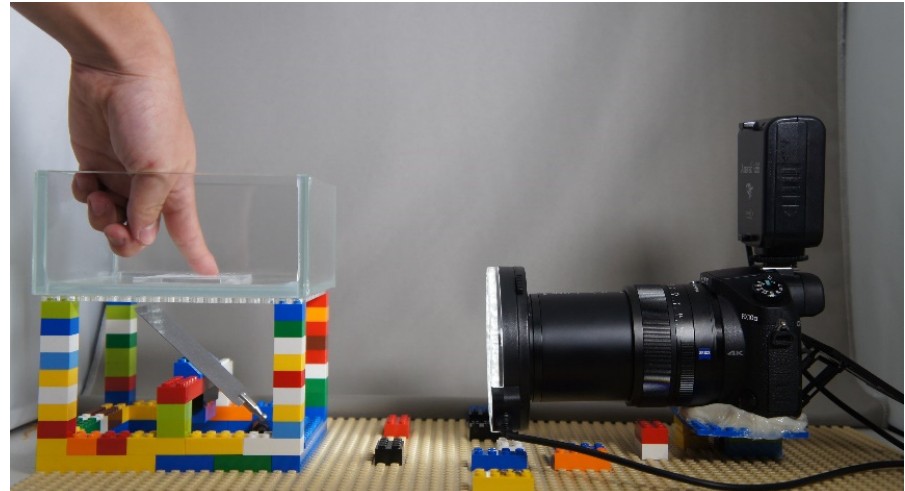

**Figure 4.** View of the measurement system taken from the side. To verify the principle of the system, we implemented a measurement system using Lego blocks. The texture is made optically transparent by filling an aquarium with a silicone oil having the same refractive index as the texture.

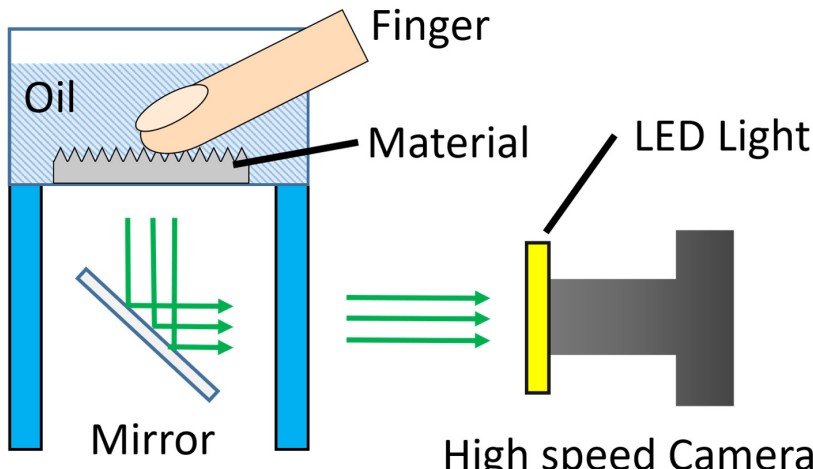

**Figure 5.** Overview of the experimental setup. By filling the water tank with oil, the texture is optically transmitted. The camera is placed in front of a mirror to observe fingertips through the textures, and the LED is attached to reduce the noise from the high-speed camera.

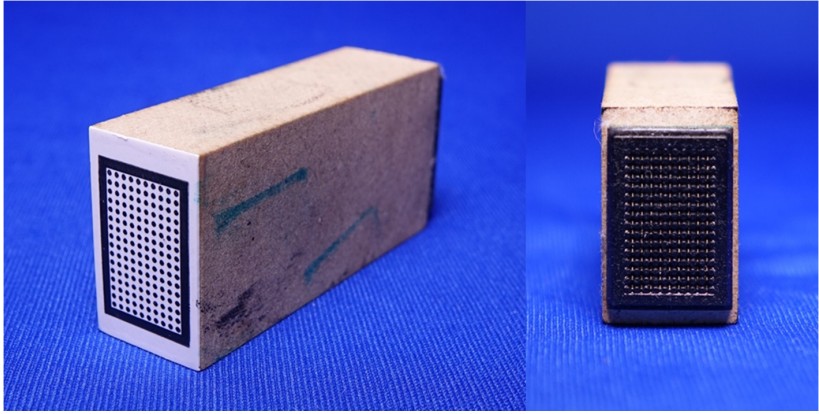

**Figure 6.** Stamp used to apply the markers to the finger. Points were placed in a 10 × 16 array, each having a 0.5 mm diameter and 1.0 mm center-to-center intervals.

### 3.3. Software

Figure 7 shows the flow of the image-processing algorithm. This software enables the tracking of the marker position applied to the fingertip. The template-matching algorithm (matchTemplate function, use TM_CCOEFF_NORMED as a similarity calculation method) from the OpenCV library (http://opencv.org) was used for position analysis.

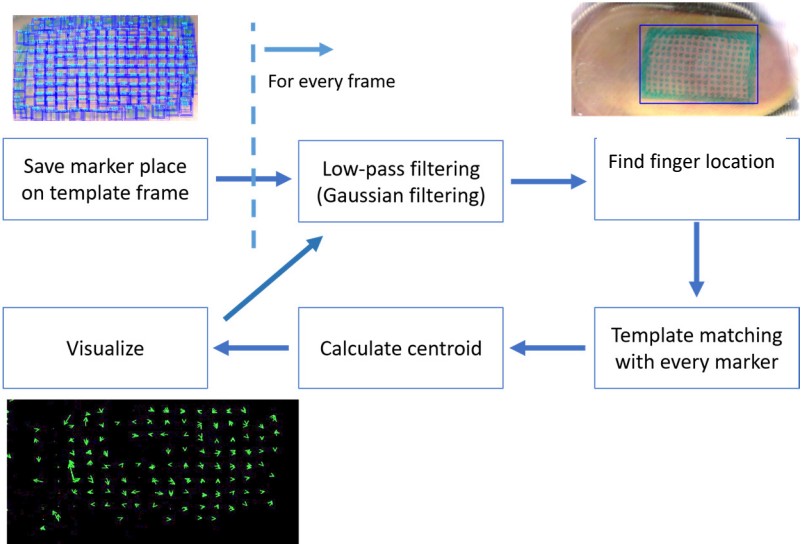

**Figure 7.** Flow of the image-processing algorithm.

The algorithm proceeds as follows. First, the two image templates, the finger template, and the marker template are manually cut out from the skin displacement images. The finger template includes each marker point and a frame surrounding the point. The marker template contains a single point marker extracted from the entire fingertip. Then, the initial position of each marker is extracted from the first frame. For this, template matching using marker templates is applied. In this case, the region having a similarity of 0.65 or more is used as a marker. If a non-marker is tracked, it is removed manually during the subsequent analysis process. Then, the image of the initial position of each marker is saved as a template for each marker. For each frame, we extract the finger position using the finger template and estimate the displacement of the marker. A Gaussian filter (window-size: 11 pixels) is applied to the frame as a preprocessor. The template matching for each marker is performed by cropping a region with a 5-pixel offset to the marker region in the previous frame from the new frame. This is done to improve the computational speed and prevent the tracking of different markers. The centroid position is calculated from the similarity results of template matching. This result is considered to be an estimated marker position. This makes it possible to calculate the displacement from the previous frame with sub-pixel accuracy. This process is applied to the video (Supplementary Materials Videos S1 and S2). The measured results are output in the form of x-axis displacement (in the slope direction) and y-axis displacement (perpendicular to the slope direction). The movement of each marker is visualized using arrows.

## 4. Experiments Evaluation

To evaluate the accuracy of the system, an evaluation was performed using a printed marker as an ideal condition.

### 4.1. Methods

The marker image printed on oil-proof paper was immersed, and marker displacements were measured using the same procedure as that for the actual measurements. The artificial finger with

markers is shown in Figure 8. The markers were printed with the same dimensions as the markers to be applied to the finger (see Section 3.2). In this experiment, the movement of each marker should not be observed, because there is no skin displacement. Marker movement would be attributed to the noise of the measurement system because of the refractive index mismatch, pixel discretization, and the marker-tracking algorithm.

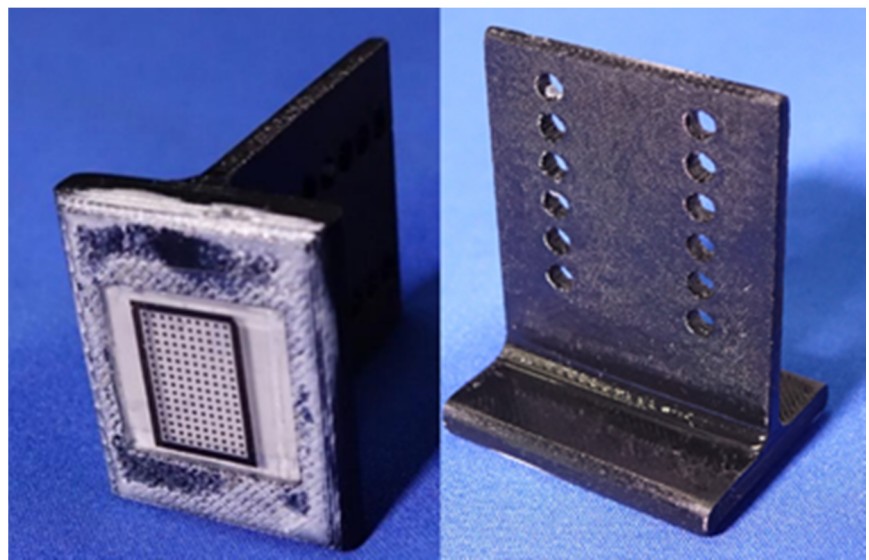

**Figure 8.** Artificial finger with markers printed on oil-proof paper.

Figure 9 shows the surface used for the experiments, which have hairline textures. The surface was prepared by processing an acrylic plate using a laser cutter. The size of the textured surface was $100 \times 50 \times 3$ mm, and the width of the textured portion was 75 mm. The pitch width was 0.6 mm.

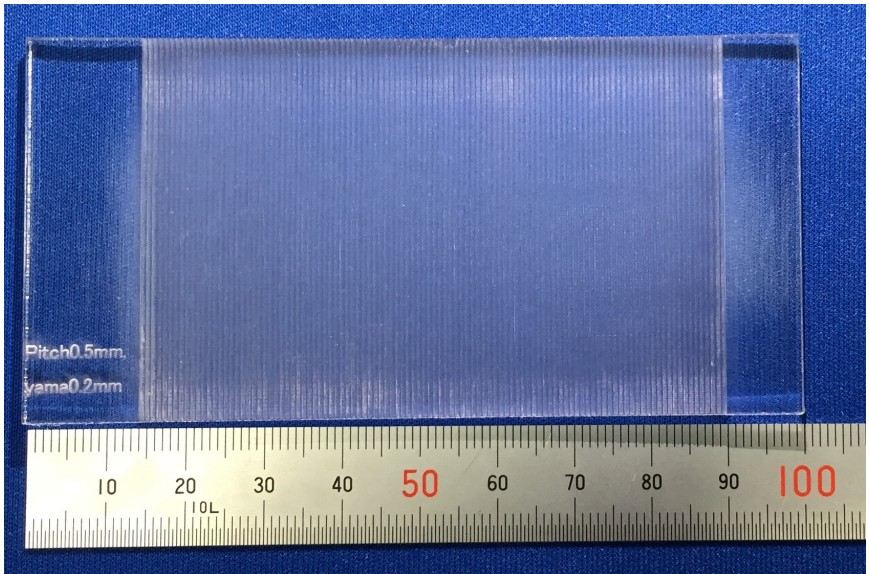

**Figure 9.** Textured surface used for the experiment. Pitch width: 0.6 mm.

A linear actuator (YAMAHA (Japan), T4L) was used to move the artificial finger. The artificial finger moved 50 mm to the left and right on the textured surface at approximately 1.3 Hz. The artificial finger was pressed from the top so that there was no space between the marker and the textured surface.

### 4.2. Results

Figure 10 shows the marker recognition results. The red squares in the figure show the locations of the recognized markers. This result indicates that all the markers were recognized. Figure 11 shows the displacement of the No. 0 marker in the horizontal direction. The average peak-to-peak displacement in each cycle was approximately 20 μm. This indicates that there was a recognition error of approximately 20 μm at this marker. After performing the same calculation for all the markers, it was found that there was an average error of approximately 31 μm. Therefore, the measurement system can effectively function as a skin displacement measurement tool for displacements greater than 31 μm.

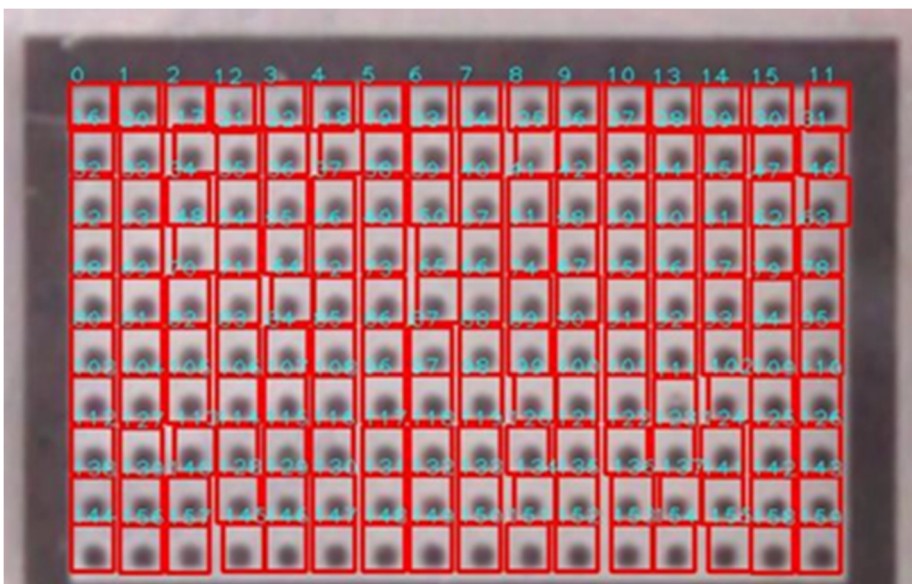

**Figure 10.** The result of marker recognition. The image is blurred because the Gaussian filter has been applied. A red square indicates that the program recognized that area as a marker. A video of the the measurement video (Video S1) is presented in Supplementary Materials.

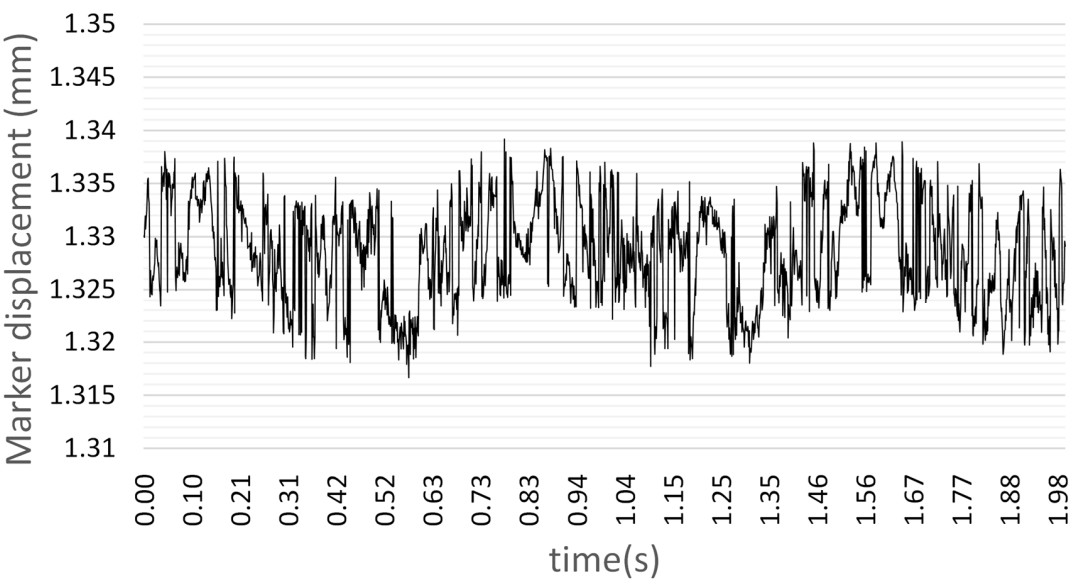

**Figure 11.** Displacement of the No. 0 marker in the slope direction from Figure 10. The variation generated by this measurement is noise. This is because there is no variation between the markers in this measurement. It can be seen that the average noise level in this marker is 20 μm.

## 5. Discussion

In this paper, a measurement system for direct skin displacement on a textured surface was proposed. To verify the validity of this method, the degree of optical misalignment was verified via simulation, and the measurement accuracy was verified using the implemented measurement system. A simulation to estimate the refractive index change and the optical misalignment caused by the texture angle revealed that this system can measure skin displacement directly related to the SAI and RAI activities on the textured surface. Previous measurements of skin displacement on textured surfaces have mainly been made on PCs having wide receptive fields. It is noteworthy that the firing thresholds of the SAI and RAI can now be measured under appropriate conditions. However, it is difficult to measure high-frequency (>200 Hz) vibrations with an amplitude of less than 1 μm, which leads to the activity of the PCs in this system. One way to solve this problem is to attach an acceleration sensor to the nail [2] or a device that can measure the vibration propagated from the fingertip to the middle of the finger in a high-frequency band [26].

The experimental results suggest that skin displacement on a textured surface can be measured in oil with a noise level of approximately 30 μm. The results are less than the ideal values obtained by the simulation. Although this value is better than the firing thresholds of the SAI, the accuracy is below the firing thresholds of the RAI (18 μm (4 Hz) at maximum and 0.7 μm (50 Hz) at minimum [27]). The present results are inferior in accuracy to those of existing systems used for tactile measurements (e.g., accelerometers, laser Doppler sensors). However, with an accuracy below the firing threshold of the tactile receptors, our system can measure skin deformation on a texture, and also observe it spatially and temporally. These results satisfy the requirements for reproduction on a tactile display.

There are three possible reasons for the low measurement accuracy in this study: (i) the wide angle of view, (ii) the marker-tracking system, and (iii) hardware implementation. For (i), the current measurement system requires a wide angle of view, because the finger traces the texture, which results in poor per-pixel resolution. To solve this problem, the resolution of the camera should be improved or the texture should be moved by fixing the fingertips. In the latter case, the fingertips can be measured up to the very edge of the angle of view when the finger is not moved, resulting in a lower millimeter-per-pixel value and a higher resolution. However, this may have some effect on roughness perception, because it would be a passive touch instead of an active touch. For (ii), the software can be improved to reduce noise, including a more robust calculation of finger location with adaptive filtering, because the marker location is subtracted from the finger location, and the error of the finger location effects all markers. Additionally, the accuracy of the marker displacement estimation can be improved by improving the resolution of the finger fixation as mentioned in the solution of (i) because, in the current setup, one pixel corresponds to 30 μm. For (iii), because we used Lego in this hardware implementation, distortions were generated while moving the artificial fingers. To solve this problem, it will be necessary to use more rigid materials such as aluminum frames in the future.

A possible application of this research is the measurement of the spatiotemporal displacement of skin displacement on a variety of natural textures. The proposed measurement system enables us to analyze texture clustering using spatiotemporal information, whereas previous studies used vibration measurements from accelerometers, etc. Furthermore, the displacement data obtained here will enable us to study the vibration presentation algorithm on a tactile display. Another possible application is the measurement of skin displacements in areas other than the fingertips.

## 6. Conclusions

In this study, we developed a system to measure skin displacement on textured surfaces. In the developed system, the texture plate is immersed in silicone oil having almost the same refractive index as acrylic, which makes the texture optically invisible. A finger with optical markers is used to trace the plate, and each marker movement is obtained using a high-speed camera and an image-processing algorithm. The tactile sensation during measurement in oil can be subjectively

evaluated, and the relationship between skin displacement and the tactile sensation can be clarified by using our measurement system.

The next step is to select a marker that provides a clearer image to the fingertips to reduce the measurement noise and optically magnify the scene by fixing the finger and moving the plate. We also plan to use other textures in future studies to further observe and measure skin behavior.

**Supplementary Materials:** The following are available online at http://www.mdpi.com/2076-3417/10/12/4184/s1, Video S1: Video of marker recognition and measurement activity. Each frame is trimmed at marker area. Video S2: Video of marker area recognition and measurement activity.

**Author Contributions:** Conceptualization, S.K. and H.K.; Methodology, H.K.; Software, S.K.; Validation, S.K. and H.K.; Writing—Original Draft Preparation, S.K. and H.K.; Writing—Review and Editing, S.K. and H.K.; Project Administration, H.K. All authors have read and agreed to the published version of the manuscript.

**Funding:** This work was funded by JSPS KAKENHI, grant number 15H05923 (Grant-in-Aid for Scientific Research on Innovative Areas, "Innovative SHITSUKSAN Science and Technology").

**Conflicts of Interest:** The authors declare no conflict of interest.

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
