# Peer review of "Measurement System for Finger Skin Displacement on a Textured Surface Using Index Matching"

_applsci, doi:10.3390/app10124184_

Round 1
Reviewer 1 Report
The paper discusses the design and evaluation of a system to measure skin deformation on textured surfaces. The subject of the paper, evaluation, and findings are all very important to the intended population of the journal. The methods, evaluation, and contribution are worth to publish.
Other comments:
- Provide abbreviations for SAI, RAI, FAI, and PC the first time they are used.
- The authors briefly touched upon the limitations of the proposed method but have not discussed the influence /impact of the two limitations. Please elaborate on this.
- Good summary on the literature review
- Citation error on multiple instances
- Provide descriptive captions for figures 4 and 5
- Not sure how fixing the finger position (& moving the plate) would increase the accuracy in measurement. Please explain.
Reviewer 2 Report
The paper deals with a measurement system of finger skin displacement on a texture surface using index matching.
The topic of the paper is interesting to the reader. However, the overall quality of the manuscript is low and should be deeply improved before considering the paper suitable for publication.
The main contribution and novelties of the work with respect to the previous literature are not clear. The authors should clearly state and summarize which are the novel contributions. It is not clear which is the gap that the authors would like to fill.
The description of the software is too simple. It would be impossible to the reader to reproduce the experimental tests. I suggest to deeper describe the software and the parameters of the algorithms used in the analysis.
The discussion of the results is very poor and should be improved.
Furthermore, there are many typos in the manuscript and English should be improved.
The quality of the figures is very low. I suggest to decrease the size of figures 1, 2, 4, 5, 6, 8, 9.
The quality of figures 3, 10 and 11 is not acceptable. These figures should be replaced with high quality figures in resolution.
Figure 11 is not clear and should be replaced and better commented in the text.
The authors state that the proposed device could be useful to better investigate the displacement of the skin subjected to vibrations. The authors should better describe this possible application with respect to real scenarios, such as the displacement of the skin when in contact with vibrating motors. It would be interesting to know if the proposed device could be used only for fingertips or to other parts of the hand as well. I suggest the authors to add the following reference:
Scalera, L., Seriani, S., Gallina, P., Di Luca, M., & Gasparetto, A. (2018). An experimental setup to test dual-joystick directional responses to vibrotactile stimuli. IEEE Transactions on Haptics, 11(3), 378-387.
Minor remarks:
What do SAI and RAI mean? The acronyms have to be explained before used.
What does “PCs with low spatial resolution” in line 38 mean? Please explain.
What do SA and RA mean in lines 70-71? Please explain.
Line 161 Error! Reference source not found.6). Please correct.
Please indicate where the liquid is in Figure 5.
Line 196: Error! Reference source not found.8. Please correct.
Reviewer 3 Report
The authors have improved their paper by adding some corrections proposed by previous reviewers.
Although these changes have improved the quality of the paper and clarify their contributions, some sentences should be improved and some points should be clarified:
- "from the perspective of both space-time and space-time" --> Repetition
- "The camera is optically placed on the bottom of the image using a mirror to observe fingertips through textures" --> The camera is not on the botton of the image. The image is generated inside the camera.
- The quality of Fig.3 and 7 should be improved. They are too blurred.
- In Fig.7 and in the text explaining it, the authors talk about "point of gravity". I imagine they mean "centroid". They should be more precisely and use the correct term.
- The test methodology should be better explained: "A linear actuator (YAMAHA, T4L) was used to move the model finger. The model finger moved 50 mm to the left and right on the textured surface at approximately 1.3 Hz." Does the linear actuator moves all the body of the human or do you cut the finger of the human in order to move it with your actuator?
- How does the frequency of the finger motion (at 1.3Hz) is related with the firing thresholds of RAI, SAI and PCs and the frames per second recorded by the camera?
Round 2
Reviewer 2 Report
The paper was not sufficiently improved with respect to the previous version and, therefore, it cannot be considered acceptable for publication.
1) The main contributions of the paper with respect to the present literature are still not clear:
- In the literature there are several methods for observing skin deformation. What is the novel contribution of this method?
- What kind of "optical simulations" did the authors perform?
- How the authors verified the accuracy of the equipment? The experimental setup is made of Lego bricks which is not professional and scientifically sound. The deformations in the structures have not been taken into account.
2) The discussion of the results is still not clear. The originality of the results is not properly highlighted.
3) The quality of all the figures of the paper is too low for a journal publication. Figures are blurred and data cannot be appreciated. Figure 3 is completely blurred. What should the reader understand from Figure 10? It is completely useless.
4) It would be interesting to know if the proposed device could be used only for fingertips or to other parts of the hand as well. The authors did not properly answer this question.
5) English is still very poor and in many sentences the language is not suitable for a scientific paper.
Round 3
Reviewer 2 Report
The paper has been improved with respect to the previous version.